# Natural Medicines and Their Underlying Mechanisms of Prevention and Recovery from Amyloid Β-Induced Axonal Degeneration in Alzheimer’s Disease

**DOI:** 10.3390/ijms21134665

**Published:** 2020-06-30

**Authors:** Tomoharu Kuboyama, Ximeng Yang, Chihiro Tohda

**Affiliations:** 1Section of Neuromedical Science, Institute of Natural Medicine, University of Toyama, Sugitani 2630, Toyama 930-0194, Japan; t-kuboyama@daiichi-cps.ac.jp (T.K.); ximeng@inm.u-toyama.ac.jp (X.Y.); 2Laboratory of Pharmacognosy, Daiichi University of Pharmacy, 22-1 Tamagawa-cho, Minami-ku, Fukuoka 815-8511, Japan

**Keywords:** amyloid β, axon, traditional medicines, Polygalae Radix, diosgenin, naringenin, kihito

## Abstract

In Alzheimer’s disease (AD), amyloid β (Aβ) induces axonal degeneration, neuronal network disruption, and memory impairment. Although many candidate drugs to reduce Aβ have been clinically investigated, they failed to recover the memory function in AD patients. Reportedly, Aβ deposition occurred before the onset of AD. Once neuronal networks were disrupted by Aβ, they could hardly be recovered. Therefore, we speculated that only removal of Aβ was not enough for AD therapy, and prevention and recovery from neuronal network disruption were also needed. This review describes the challenges related to the condition of axons for AD therapy. We established novel in vitro models of Aβ-induced axonal degeneration. Using these models, we found that several traditional medicines and their constituents prevented or helped recover from Aβ-induced axonal degeneration. These drugs also prevented or helped recover from memory impairment in in vivo models of AD. One of these drugs ameliorated memory decline in AD patients in a clinical study. These results indicate that prevention and recovery from axonal degeneration are possible strategies for AD therapy.

## 1. Introduction

Dementia is one of serious refractory diseases in the world, especially in aging societies such as Japan, Europe, and United States. Alzheimer’s disease (AD) is the most frequent cause of dementia. Estimated 35 million people were suffered from AD in 2017 [1]. AD is a progressive and irreversible neurodegenerative disease. Not only is the quality of life in AD patients extremely low, but oftentimes, the family bears a heavy burden of care for the patient. The mean simulated survival of AD patients is 19.0 years [2]. Amyloid β (Aβ) is a critical cause of AD [3,4,5,6]. Aβ is cleaved from amyloid precursor protein (APP) by processing enzymes β-secretase and γ-secretase [7]. There are two types of AD, namely familial and sporadic. In both cases, Aβ induces neurodegeneration and memory deficits [5,7]. In familial AD, mutations in *APP* or presenilin (PS), a component of γ-secretase, result in Aβ overproduction [7]. In sporadic AD, other gene mutations such as mutations in *APOE* and *TREM2*, are possibly related to Aβ accumulation [7].

Recently, many candidates targeting Aβ reduction such as β-secretase and γ-secretase inhibitors and anti-Aβ antibodies have been clinically investigated for AD therapy. Unfortunately, these candidates have not been clinically approved, as most of them failed to achieve positive effects in primary outcomes [8,9]. Reportedly, a considerable amount of Aβ had already been accumulated in the brain at the onset of AD [10]. Therefore, prevention of Aβ production and removal of Aβ might not be adequate to treat AD patients. Currently, only four medicines are available for AD therapy, namely donepezil, galantamine, rivastigmine, and memantine, which regulate the synaptic function to improve memory. Although these drugs are effective for AD, their effects are limited. These drugs only delay deterioration of AD symptoms, but do not help patients recover from AD. Moreover, they show limited benefits when compared with their adverse effects [11,12,13,14]. Therefore, the French minister of health delisted these medicines for AD therapy in 2018 [15]. In AD, Aβ induces irreversible and fundamental neural atrophy in the brain [16,17,18], leading to severe memory deficits. Therefore, drugs that only modulate the synaptic function might not be adequate to cure AD. Structural degeneration of neuronal networks are proceeded and promoted by Aβ.

At the onset of AD, neurodegeneration and memory deficits are not severe [10]. Therefore, we hypothesized that if Aβ-induced neural atrophy is prevented or recovered, memory function might be maintained or restored, respectively. Aβ induces axonal degeneration and neuronal death in vitro and in the brain of AD patients [16,17,19]. We established new in vitro models of Aβ-induced irreversible axonal degeneration [20,21,22]. The present review summarizes our challenges in proving our hypothesis using these models. We focused especially on natural medicines and investigated their effects using the in vitro AD models, in vivo AD models, and human subjects. Our study proposes new strategies for the prevention and treatment of AD by protecting against and recovering from Aβ-induced axonal degeneration.

## 2. Prevention of Aβ-Induced Axonal Degeneration

Aβ induces neuritic degeneration in in vitro models, in vivo mouse models, and AD patients, leading to disruption of neuronal networks and memory deficits [17,21,23,24,25,26,27,28]. Although several studies have suggested a variety of mechanisms of Aβ-induced neuritic degeneration, such as abnormality of autophagy and activation of calcineurin [29,30], its critical cause has not been identified. We observed Aβ-induced acute collapse of axonal growth cone within 1 h in cultured neurons [20,31]. This culture system made it possible to visualize and analyze early events in the Aβ-induced axonal degeneration process. Various inhibitors and mutated gene transfection revealed that Aβ induced Ca^2+^ signaling via N-methyl-D-aspartate receptor and transient receptor potential channel, activated calcineurin and calpain, facilitated clathrin-mediated endocytosis, and induced growth cone collapse (Figure 1). In case of repulsive axon guidance, asymmetric clathrin-mediated endocytosis removes the plasma membrane and β1-integrin from the surface of the growth cone, leading to turning the growth cone away from the reduced membrane [32]. Therefore, Aβ might remove some functional molecules from the surface of the growth cone, leading to the collapse of the growth cone and then causing atrophy of axons.

Intracerebroventricular (i.c.v.) injection of Aβ to adult mice induced axonal degeneration in the brain and memory deficits [33,34,35]. When specific inhibitors of clathrin-mediated endocytosis, namely pitstop 2 and myristoylated dynamin inhibitory peptide, were simultaneously i.c.v. injected with Aβ, axonal degeneration and memory deficits were prevented [31]. Simultaneous treatments with Aβ and the inhibitors prevented growth cone collapse and axonal atrophy in cultured neurons. These results show that inhibition of clathrin-mediated endocytosis prevents not only the early events elicited by Aβ such as collapse and degeneration of axons, but also the late events elicited by Aβ such as memory deficits. Clathrin-mediated endocytosis is a novel and critical target for AD prevention.

5XFAD mouse is a transgenic model of AD, which overexpresses familial AD mutations, namely K670N/M671L (Swedish), I716V (Florida), and V717I (London) mutations in APP and M146L/L286V mutations in PS1, and shows Aβ deposition, phosphorylated tau deposition, neuronal loss, axonal degeneration, and memory deficits [36,37,38,39]. In the brain of 4-month-old 5XFAD mice, Aβ plaques and degenerated axons were few in number and novel object recognition memory was normal [40]. However, in 5-month-old mice, Aβ plaques and degenerated axons were significantly increased and novel object recognition memory was impaired. We found that water extract of Polygalae Radix (PR) (roots of *Polygala tenuifolia*, a traditional herbal medicine) inhibited Aβ-induced endocytosis in growth cones in cultured neurons [40]. When PR was orally administered to 4-month-old 5XFAD mice for 56 days, degenerated axons were significantly decreased in number when compared with that in vehicle-administered mice. Object recognition memory was not impaired, but there was no change in the increase in Aβ plaques after PR administration. In cultured neurons, PR prevented Aβ-induced growth cone collapse even in the presence of Aβ. These findings indicate that PR prevented Aβ-induced toxicity even in the presence of Aβ possibly via inhibition of endocytosis. Another group also showed that administration of PR extract to 4-month-old 5XFAD mice for 2 months prevented memory deficits [41]. The authors showed neuroprotective effects of the PR extract against Aβ. Tenuifolin, a constituent in PR, counteracted Aβ toxicity in in vitro and in vivo [42]. PR is traditionally used for memory loss in East Asia [43,44]. PR has been shown to enhance cognitive functions in healthy elderly individuals [45]. Therefore, PR is a promising preventive candidate drug for AD.

## 3. Recovery from Aβ-Induced Axonal Degeneration

### 3.1. Ashwagandha

Ashwagandha (a root of *Withania somnifera* Dunal), also called Indian Ginseng, is one of the most important herbal drugs in Indian traditional Ayurvedic medicine; namely, as a rasayana drug that is used for longevity and increasing vital energy and intellectual power. Ashwagandha is clinically applied for dyspepsia, anxiety, depression, insomnia, and loss of memory. Various pharmacological studies of Ashwagandha and its constituents have been investigated, such as anti-inflammation [46,47], anti-stress [48,49], and neuroprotection [50,51]. These facts stimulated our curiosity about effects of Ashwagandha and its constituents on AD. We found that a methanol extract of Ashwagandha induced neurite outgrowth in cultured human neuroblastoma SK-N-SH cells [52]. Then, six compounds were isolated from the methanol extract as active constituents showing neurite outgrowth activities [53,54].

Effects of the active constituents of Ashwagandha in Aβ-induced degenerated condition were investigated. Aβ induces axonal atrophy, and the atrophied axons never recover, even after removal of Aβ in cultured cortical neurons [21,22], namely, Aβ induced irreversible axonal atrophy. Among the active constituents, withanolide A, withanoside IV, and withanoside VI (Figure 2) were administered to cultured cortical neurons after Aβ induced axonal atrophy. As a result, these three constituents induced axonal growth in the Aβ-treated neuron culture [21,55,56]. These three constituents were orally administered to the Aβ-i.c.v.-injected mice after memory deficits occurred. They ameliorated degenerated axons in the brain and recovered memory. Among these three constituents, withanoside IV was shown to be metabolized into sominone (Figure 2) after oral administration [55]. Sominone induced axonal growth in Aβ-treated cultured neurons and recovered degenerated axons in the brain and memory in 5XFAD mice at 6–8 months of age when the mice already showed axonal degeneration and memory deficits. These results indicate that sominone is an active principle of withanoside IV after oral administration. Sominone induced axonal growth in normal cultured neurons and in the brain of normal adult mice [57]. Sominone also enhanced spatial memory in normal adult mice. In those cases, sominone increased RET phosphorylation, which meant activation of RET. RET is a receptor of the glial cell line-derived neurotrophic factor (GDNF). In cultured normal neurons, GDNF induced axonal growth, knocking down RET diminished sominone-induced axonal growth, while sominone enhanced the secretion of GDNF. These results indicate that sominone induces axonal growth via GDNF-RET signaling and enhances memory in normal adult mice. Considering that transplantation of GDNF-overexpressing cells ameliorated memory in 5XFAD mice [58], withanoside IV and sominone might improve memory via GDNF-RET signaling in 5XFAD mice. We synthesized a novel compound, denosomin (Figure 2), which is a derivative of sominone. Denosomin induced axonal growth in Aβ-treated neuron culture as much as sominone did [59]. Denosomin showed stronger neuroprotective effects against Aβ than sominone at low concentrations. Therefore, denosomin is expected to be a novel anti-AD drug.

Withanoside IV and denosomin recovered motor function in spinal cord injured mice [60,61]. Axonal growth activities of these compounds might contribute to the recovery from spinal cord injury. Ashwagandha and its constituents were reportedly effective for various neurological disorder models besides AD and spinal cord injury, such as Parkinson’s disease, addiction, anxiety, schizophrenia, dyslexia, depression, and so on [62]. Ashwagandha can be a source of novel drugs for not only AD, but also other various refractory neurological disorders.

### 3.2. Diosgenin

Diosgenin is a steroid sapogenin derived from Dioscorea wild yam. Rhizome of *Dioscorea japonica* Thunberg is used as a tonic agent for aging people in Japanese traditional Kampo medicine. We screened traditional medicine-derived compounds and found diosgenin to be a potent stimulator of axonal growth. Therefore, we postulated that diosgenin had anti-AD activities.

Treatment with diosgenin significantly induced axonal growth after axons were already atrophied by Aβ in cultured neurons [63]. Diosgenin was administered for 20 days to 6- to 8-month-old 5XFAD mice when axonal degeneration and memory impairment had already occurred. Diosgenin significantly improved the degenerated axons and memory function. These results indicated that diosgenin normalized degenerated axons, induced axonal growth, helped in reconstruction of neuronal networks, and improved memory in 5XFAD mice.

To clarify the mechanisms of diosgenin, a protein capable of direct binding with diosgenin was identified using the drug affinity responsive target stability (DARTS) method [64]. Briefly, diosgenin was mixed with neuron lysate and was treated with a protease. During this reaction, diosgenin binds to a target protein, leading to conformational modification of the protein. Vulnerability of the protein against the protease is possibly changed. After poly-acrylamide gel electrophoresis, band intensity of the target protein should be different between drug treatment and control treatment. The changed band is considered a candidate for direct binding with diosgenin. Using the DARTS method, various proteins that can directly bind with compounds or molecules have been identified [65,66]. In case of diosgenin, 1,25D_3_-membrane-associated rapid response, steroid-binding protein (1,25D_3_-MARRS) was identified, which is a membrane receptor for 1α,25-dyhydroxyvitamin D_3_ [67]. Direct binding of diosgenin with 1,25D_3_-MARRS was indicated by several other experiments, such as docking simulation of diosgenin to 1,25D_3_-MARRS. By knocking down 1,25D_3_-MARRS in cultured neurons, axonal growth activity of diosgenin was completely blocked. Treatment with a functional blocking antibody for 1,25D_3_-MARRS also inhibited diosgenin-induced axonal growth. By using kinase inhibitors, it was suggested that diosgenin induced axonal growth via phosphoinositide 3-kinase, extracellular signal-regulated kinase, protein kinase C, and protein kinase A. Since these kinases are reported to be involved in 1,25D_3_-MARRS signaling [67,68,69,70], diosgenin probably induced axonal growth via these kinases under 1,25D_3_-MARRS signaling (Figure 3).

In normal cultured neurons, diosgenin induced axonal growth [73]. In normal young mice, diosgenin increased axonal density in the brain cortex and enhanced memory. These in vitro and in vivo effects of diosgenin were completely blocked by the functional blocking antibody for 1,25D_3_-MARRS. These results indicate that 1,25D_3_-MARRS is a receptor for diosgenin with positive effects on axons and memory in AD-like conditions as well as in normal conditions.

To further investigate the downstream signaling of diosgenin, gene expression in the brain cortex of diosgenin-administered 5XFAD mice was comprehensively analyzed [71]. Expression of heat shock cognate 70 (HSC70) was decreased by diosgenin administration. On the other hand, HSC70 was increased in the brain cortex of 5XFAD mice when compared with HSC70 expression in wild-type mice. In cultured neurons, Aβ treatment increased HSC70 expression, while diosgenin-treatment decreased it. This in vitro effect of diosgenin was canceled by the functional blocking antibody for 1,25D_3_-MARRS. Knocking down of HSC70 alone induced axonal growth in cultured neurons. VER-155008, an inhibitor of HSC70, also induced axonal growth in the Aβ-treated neuron culture [72]. VER-155008 attenuated degenerated axons and improved memory in 5XFAD mice. Diosgenin decreased Aβ plaques and phosphorylated tau in the brain of 5XFAD mice [63]. Phosphorylated tau is another cause of AD, which is accumulated by Aβ-induced activation of glycogen synthase kinase 3β, Cyclin dependent kinase-5, and so on, leading to microtubule disassembly and neuritic degeneration [74,75,76]. VER-155008 also decreased Aβ plaques and phosphorylated tau in the brain of 5XFAD mice [72]. These results indicate that diosgenin binds to 1,25D_3_-MARRS and downregulates HSC70, leading to attenuation of degenerated axons, decrease in Aβ plaques and phosphorylated tau, and recovery from memory deficits in 5XFAD mice (Figure 2). We speculated that HSC70 promoted α-tubulin degradation in axons and diosgenin attenuated the degenerated axons via downregulating HSC70 and retaining α-tubulin expression in axons [71]. Further investigations are needed to clarify how 1,25D_3_-MARRS signaling downregulates HSC70 and how HSC70 decreases Aβ plaques, decreases phosphorylated tau, and attenuates degenerated axons.

We also investigated the effects of diosgenin in humans [77]. In the investigation, diosgenin-rich yam extract was used. Healthy men and women (mean age was 46.50 years) were randomly divided into two groups. One group took the extract, and the other group took placebo for 12 weeks; then, memory function was assessed by Repeatable Battery for the Assessment of Neuropsychological Status (RBANS). After a 6-week interval period, the group previously taken placebo took the extract, and the other group took placebo for 12 weeks; then, memory function was assessed in the same way. This was a placebo-controlled, randomized, double-blind, and crossover study. As a result, the total score of RBANS was significantly increased by taking the extract. The diosgenin-rich yam extract enhanced cognitive function in healthy adult humans. This result as well as the results obtained in mice, in which diosgenin enhanced memory in both normal adult mice and 5XFAD mice [63,73], suggest that the diosgenin-rich yam extract might be effective for AD patients. We are planning to perform a clinical study for AD patients using this yam extract.

### 3.3. Drynariae Rhizoma

Drynariae Rhizoma (DR) is a rhizome of *Drynaria fortunei* (Kunze ex Mett.) J. Sm. (Polypodiaceae), which is a Chinese herbal drug used for bone fracture and for tonifying the kidneys [78]. We found that water extract of DR induced axonal growth in the Aβ-treated neuron culture [79]. Oral administration of DR extract attenuated degenerated axons and improved memory in 5XFAD mice [80]. In vitro bioassay-guided isolation is usually performed to identify the active compounds in herbal medicines [81]. However, this method may sometimes lead to the misidentification of the real active compounds, as the metabolization process in the body and permeability of the blood-brain barrier are ignored. Therefore, we aimed to identify the real active compounds by detecting compounds in the brain of 5XFAD mice after oral administration of DR extract (in vivo screening) [80]. Naringenin was identified as a compound that crossed the blood-brain barrier. Naringenin showed axonal growth activity in the Aβ-treated neuron culture, ameliorated degenerated axons, and improved memory in 5XFAD mice. The DR extract contained a low amount of Naringenin itself. However, it contained an abundant amount of naringin, a glycoside of naringenin [80]. After oral administration of the DR extract, naringin is metabolized into naringenin and absorbed [82], and then likely transferred into the brain. The DARTS method showed that collapsin response mediator protein 2 (CRMP2) is a candidate of direct binding protein with naringenin. After treatment with naringenin to cultured neurons, the neuron lysate was immunoprecipitated with an anti-CRMP2 antibody. As a result, naringenin was detected from the immunoprecipitation. These results mean that naringenin directly binds to CRMP2. Knocking down of CRMP2 diminished axonal growth activity of naringenin. In Alzheimer’s disease, Aβ induced CRMP2 phosphorylation and axonal degeneration [83,84,85]. Our culture experiments also showed that Aβ increased phosphorylated CRMP2, while naringenin decreased it. These results indicate that naringin in the DR extract is metabolized into naringenin after oral administration of DR. Subsequently, naringenin is transferred into the brain. It binds to CRMP2, dephosphorylates CRMP2, ameliorates degenerated axons, and recovers memory (Figure 4).

Oral administration of DR extract and naringenin reduced brain Aβ plaques in 5XFAD mice [80]. We focused on microglia that might contribute to the effect of naringenin against Aβ plaques. Microglia are divided into at least two phenotypes: M1 and M2. Aβ increases M1 microglia that secrete pro-inflammatory cytokines, such as interferon-γ, interleukin (IL)-1β, and tumor necrosis factor-α, and induces inflammation that leads to neural damage [86]. On the other hand, M2 microglia secrete anti-inflammatory cytokines such as IL-4, IL-10, and transforming growth factor-β [87]. M2 microglia also express Aβ degradation enzymes, insulin degradation enzyme (IDE), and neprilysin [88] and reduce Aβ plaques [89]. We established a method to distinguish M1 and M2 phenotypes in cultured microglia [90]. Using this protocol, the effects of naringenin on microglia were investigated [91]. Aβ increased M1 predominance and decreased M2 predominance. In contrast, treatment with naringenin decreased M1 and increased M2. Naringenin also increased expression of IDE and neprilysin. These findings indicate that M2 polarization by naringenin contributes to the reduction of Aβ plaques in 5XFAD mice (Figure 4).

Rhizome of *Eleutherococcus senticosus* (Rupr. & Maxim.) Maxim., also known as Siberian ginseng, is used in Japanese traditional Kampo medicine as a tonic drug. Leafs of *E. senticosus* are not used for medicine, but consumed as tea, wine, and so on. We previously found that *E. senticosus* leaf extract and its constituents enhanced object recognition memory in healthy young mice [92]. The combination of DR and *E. senticosus* leaf extract has been investigated in healthy humans [93]. Healthy men and women were randomly divided into two groups, where one group (mean age was 63.0 years) took the combination, and the other group (mean age was 64.3 years) took a placebo for 12 weeks. After that, memory function was assessed by RBANS and stress responses were measured by the public health research foundation stress checklist short form. This was a placebo-controlled, randomized, and double-blind study. As a result, the combination group significantly increased the figure recall subscore of RBANS, and improved the anxiety/uncertainly score in the stress checklist. The combination of DR and *E. senticosus* leaf extract showed memory enhancing and anti-stress effects in healthy humans. It may be worthwhile to clinically investigate this combination in AD patients.

### 3.4. Kihito and Kamikihito

Kihito is a Japanese traditional Kampo formula used for insomnia, anemia, amnesia, depression, and neurosis. Kihito is composed of 12 herbal medicines, namely Ginseng Radix (roots of *Panax ginseng* C.A. Meyer), PR (roots of *Polygala tenuifolia* Willd.), Astragali Radix (roots of *Astragalus membranaceus* Bunge), Zizyphi Fructus (fruits of *Zizyphus jujube* Mill. var. *inermis* [Bunge] Rehd.), Zizyphi Spinosi Semen (seeds of *Z. jujube* Mill. var. *spinosa* [Bunge] Hu ex H.F. Chou), Angelicae Radix (roots of *Angelica acutiloba* Kitagawa), Glycyrrhizae Radix (roots of *Glycyrrhiza uralensis* Fisch. ex DC.), Atractylodis Rhizoma (rhizomes of *Atractylodes ovata* DC.), Zingiberis Rhizoma (rhizomes of *Zingiber officinale* Roscoe), Poria (sclerotium of *Poria cocos* Wolf), Saussureae Radix (roots of *Saussurea lappa* Clarke), and Longanae Arillus (pulp of *Euphoria longana* Lam.). Among these components, we reported that metabolite M1 derived from Ginseng Radix and water extract of Astragali Radix induced axonal growth in the Aβ-treated neuron culture and improved memory in AD model mice [35,94]. We have discussed previously that PR prevented axonal degeneration and memory deficits in AD model mice [40]. We also clarified that PR induced axonal growth in the Aβ-treated neuron culture [95]. In Japanese traditional Kampo medicine, clinical application of a single herbal medicine is rare, but formulae containing a combination of several herbal medicines are frequently used for treatment. Therefore, we focused on kihito as a candidate for an anti-AD formula and investigated its effects in in vitro and in vivo AD models before initiation of a clinical study. As expected, kihito induced axonal growth after Aβ-induced axonal atrophy in cultured neurons [96]. Kihito attenuated degenerated axons and memory in AD model mice i.c.v. injected with Aβ. It also increased expression of myelin basic protein (myelin marker) and synaptophysin (synaptic marker) in the brain of AD model mice. Thus, kihito probably reconstructed neuronal networks in the brain and improved memory in AD model mice.

The effects of kihito were clinically investigated in an open-label crossover study [97]. All AD patients in this investigation were recruited from the Toyama University hospital, Japan. Duration of kihito administration was 16 weeks. The Japanese version of the Mini-Mental State Examination (MMSE-J) was used for the evaluation. The mean of the MMSE-J score at the start of the study was 20.5, and mean age of the participants was 71.8 years. All groups took acetylcholinesterase inhibitors as usual. The MMSE-J scores significantly increased during the kihito administration period when compared with the period with no kihito administration. The temporal orientation subtest score in MMSE-J was significantly improved by the kihito administration. These findings showed that kihito improved cognitive function in AD patients.

Kamikihito is a modified formula of kihito, containing kihito, Bupleuri Radix (roots of *Bupleurum falcatum* L.), and Gardeniae Fructus (fruits of *Gardenia jasminoides* Ellis). We found that kamikihito also induced axonal growth in the Aβ-treated neuron culture and induced improvement in degenerated axons and memory in 5XFAD mice [39,98]. Using the DARTS method, cytosolic aspartate aminotransferase (cAST) was identified as the direct binding protein candidate of kamikihito [99]. Knock-down and pharmacological inhibition of cAST diminished the axonal growth activity of kamikihito. Treatment with recombinant cAST induced axonal growth in the normal neuron culture and in the brains of normal adult mice and enhanced memory function in normal adult mice [100]. Thus, kamikihito induces axonal growth via cAST activation, enhances memory function, and possibly improves memory in 5XFAD mice.

## 4. Conclusions

We developed two in vitro models of AD, namely the Aβ-induced axonal growth cone collapse model and the Aβ-induced axonal atrophy model. Using these models, we found several drugs derived from natural medicines, which prevented growth cone collapse or enhanced axonal growth. These agents prevented or induced recovery from memory deficits in AD model mice, and some of them were observed to be effective in humans. These findings indicate that axonal growth is possibly an important target for AD therapy and our models have high predictive validity for anti-AD activity.

Clarifying molecular mechanisms of natural medicines is complicated. Natural medicines contain thousands of constituents, and each of them shows various pharmacological effects. It is very complicated to identify active compounds and molecular targets of their compounds. These have made it difficult to analyze natural medicines scientifically. We advocated to resolve these problems by a combination of in vivo screening and the DARTS method. Our strategies could comprehensively identify active compounds in natural medicines and their target molecules, allowing to gain a foothold for usages of the natural medicines based on scientific evidence.

Several target molecules of anti-AD agents were identified in our studies. Most of these molecules had not been reported as AD-related molecules. Further detailed analysis of the signaling cascade of these molecules might reveal novel and effective pathways for AD prevention and cure. Natural medicines are clinically used for thousands of years. Safety of natural medicines has already been established. Studies of natural medicine might be a shortcut for the development of novel anti-AD drugs.

## Figures and Tables

**Figure 1 ijms-21-04665-f001:**
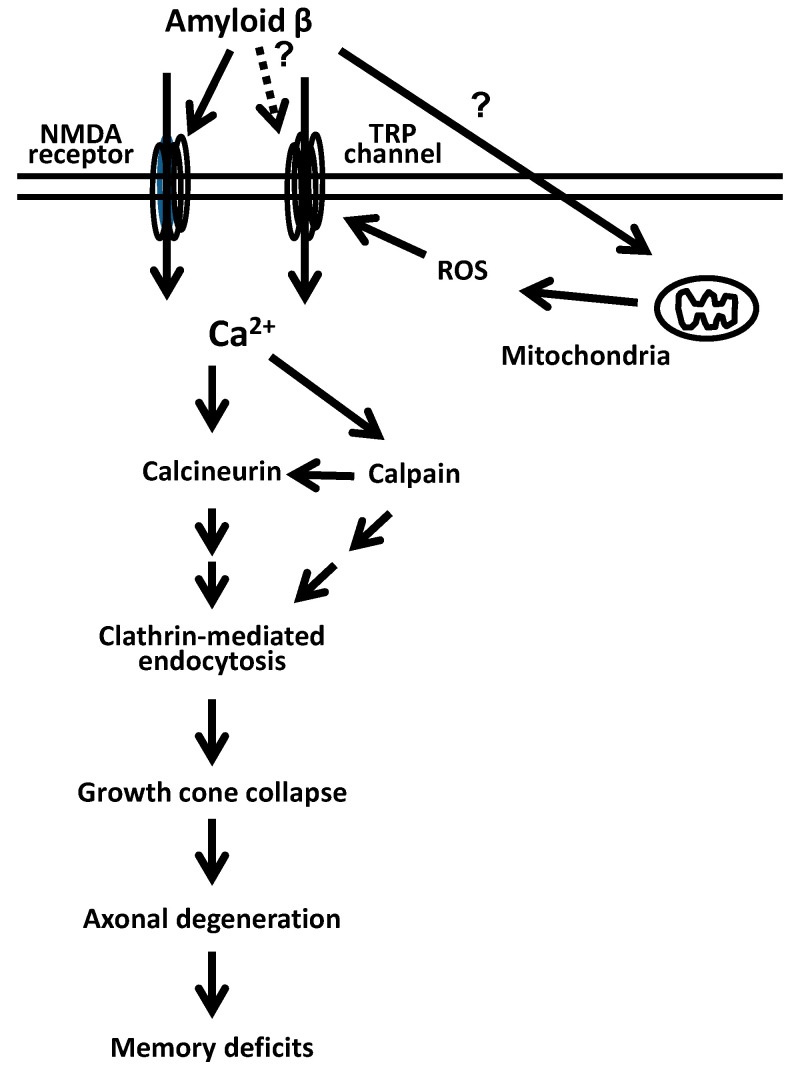
Aβ-induced growth cone collapse via clathrin-mediated endocytosis [31].

**Figure 2 ijms-21-04665-f002:**
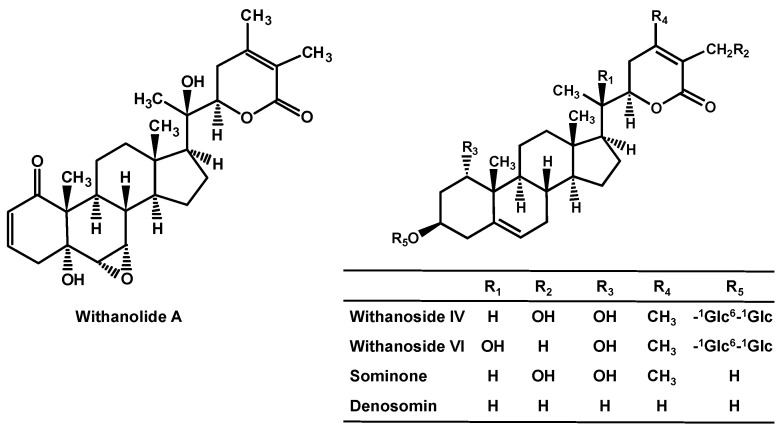
Structures of Ashwagandha-related compounds.

**Figure 3 ijms-21-04665-f003:**
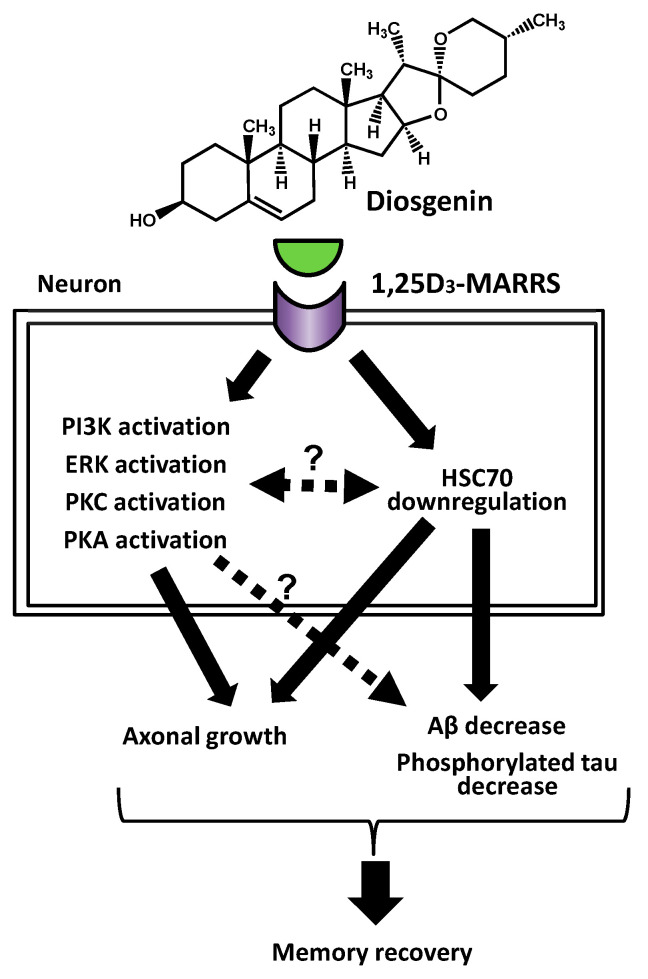
Signaling mechanism of diosgenin-induced memory recovery in Alzheimer’s disease models [63,71,72].

**Figure 4 ijms-21-04665-f004:**
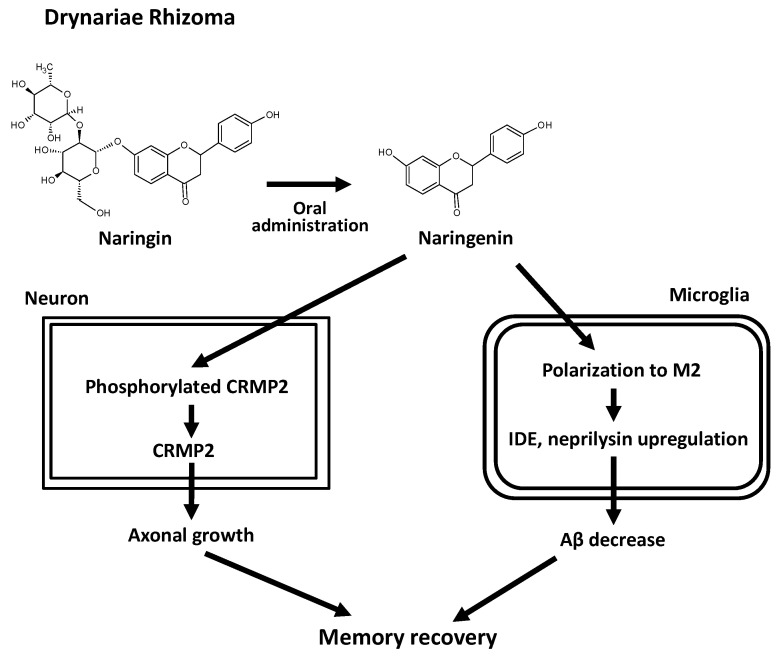
Mechanisms of memory recovery induced by Drynariae Rhizoma extract in Alzheimer’s disease models [80,91].

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
