# Peer review of "Natural Medicines and Their Underlying Mechanisms of Prevention and Recovery from Amyloid Β-Induced Axonal Degeneration in Alzheimer’s Disease"

_ijms, 2020, doi:10.3390/ijms21134665_

Round 1

Reviewer 1 Report

This is an interesting well-describe review of the effect of several traditional medicines on axonal degeneration and memory impairment in the context of AD.

Minor comments:

-All these interesting results are about molecular mechanisms of natural medicines. It could be relevant that the title of this review specifies the fact that this work focus on natural medicines.

-Figure 1: two times « axonal degeneration »

-Conclusion, line 328: change “DARTS mothod” to “DARTS method”

Author Response

Thank you for your kind suggestions.

I submit a revised manuscript as you pointed out.

Changed parts are highlighted in yellow.

Point 1: All these interesting results are about molecular mechanisms of natural medicines. It could be relevant that the title of this review specifies the fact that this work focus on natural medicines.

Response 1: I have changed as "Natural medicines and their underlying mechanisms of prevention and recovery from amyloid β-induced axonal degeneration in Alzheimer's disease".

Point 2: Figure 1: two times « axonal degeneration »

Response 2: I have deleted one of « axonal degeneration ».

Point 3: Conclusion, line 328: change “DARTS mothod” to “DARTS method”

Response 3: I have revised it. Other English mistakes have been also revised.

Reviewer 2 Report

Review by Kuboyama et al. summarizes some studies on the use of natural drugs in preventing and/or reducing axonal degeneration induced by Amyloid beta in AD. Although it is difficult to identify molecular target for traditional natural medicines (which contains several compounds), author designed and described in a comprehensive way several experiments and results. Review is well written and organized and I suggest for pubblication.

Author Response

Thank you for your efforts on checking our manuscript.

As you pointed, there were several English mistakes.

I have revised them and submit a revised manuscript.

Revised parts are shown in yellow. 

This manuscript is a resubmission of an earlier submission. The following is a list of the peer review reports and author responses from that submission.